# Exploring Interindividual Variability in Resilience to Stress: Social Support, Coping Styles, and Diurnal Cortisol in Older Adults

**DOI:** 10.3390/bs15050631

**Published:** 2025-05-06

**Authors:** Marie-Josée Richer, Sébastien Grenier, Pierrich Plusquellec

**Affiliations:** 1School of Psychoeducation, Art and Sciences Faculty, University of Montreal, Montreal, QC H2V 2S9, Canada; pierrich.plusquellec@umontreal.ca; 2Department of Psychology, Art and Sciences Faculty, University of Montreal, Montreal, QC H2V 2S9, Canada; sebastien.grenier@umontreal.ca; 3Research Center of the Institut Universitaire de Gériatrie de Montréal, Montreal, QC H3W 1W5, Canada; 4Research Center in Non-Verbal Communication Science, Research Center of the University Institute in Mental Health of Montreal, Montreal, QC H1N 3M5, Canada

**Keywords:** aging, older adults, stress, stress resilience, HPA axis, coping mechanisms

## Abstract

The psychobiological response to stress is known to be a key factor affecting health at any age, but especially in older adults. It involves the hypothalamic-pituitary-adrenal (HPA) axis, a hormonal circuit whose product is the activation of cortisol. We sought to explore the relationships leading to resilience to stress, as exemplified by the model of aging, stress, and resilience, in a sample of older adults at risk for mental health problems. Specifically, we examined the concurrent effects of individual age-related determinants, social support, and coping style on the cortisol awakening response (CAR), the cortisol area under the curve (AUC) with respect to ground, and the rate of change of cortisol from the awakening peak to bedtime. Our results showed an association between life impairment and health problems on the three indicators of HPA disturbance. An higher AUC was also observed in older age and in individuals reporting more major life events. Less use of avoidance coping was also associated with greater levels of CAR and AUC. Although significant, the measured determinant explained only a small part of the total interindividual variability in our three cortisol indices. Other factors, such as same-day stressors especially in older populations at risk for psychological distress, should be considered in future studies.

## 1. Introduction

The psychobiological response to stress, crucial for healthy aging, involves the hypothalamic-pituitary-adrenal (HPA) axis activating cortisol ([37]), with stress-induced cortisol building on circadian levels ([59]). Initially acute, stress can become chronic with aging-related persistent stressors like caring for a sick spouse, chronic illness, or the loss of functional autonomy ([5]; [31]; [36]; [44]).

In the context of chronic stress in elders, three main disturbances in the HPA axis have been reported in the scientific literature: (1) a cortisol surge in the 30 min after waking that is greater than the average 50–60%, (2) a greater diurnal cortisol secretion, and (3) a flatter diurnal cortisol slope ([18]). HPA axis dysfunction and chronic stress have been associated with a substantial list of health problems in older adults, such as diabetes ([33]) and cardiovascular disease ([54]), in addition to depressive and anxiety symptomatology ([14]; [27]).

[18] ([18]) developed a model linking resilience to stress in aging with diurnal cortisol’s role, supported by research showing age-related changes in cortisol patterns from studies on humans and animals ([24]; [47]). These changes, including higher cortisol levels, a flatter cortisol slope, and weakened feedback control, suggest that aging leads to a prolonged exposure to cortisol and reduced stress-coping capacity.

However, strong interindividual variability in diurnal cortisol levels has been observed, especially in older adults. Many individual determinants of aging have been shown to influence cortisol levels. From a life course perspective, interactions of demographic and developmental processes contribute to the accumulation of stressors over time. Being female ([22]), of lower socioeconomic status ([2]), having comorbidities ([43]), and losing functional autonomy ([45]) have been associated with changes in diurnal cortisol and ultimately contribute to greater inequalities in opportunities for healthy aging ([16]).

In addition, malleable determinants have been shown to influence physiological resilience to stress, such as social determinants and affect regulation ([18]). Smaller network size, network dissatisfaction, and a tendency not to seek emotional and instrumental help have all been associated with HPA dysfunction in aging (for reviews, see [10] ([10]) and [13] ([13])). In terms of affect, the model of aging, stress, and resilience postulates that the use of coping strategies and the ability to regulate one’s emotions are translated by the presence of more positive emotions in elders. Specifically, some studies have examined the relationship between multiple coping strategies and cortisol levels in older adults; however, the results are inconsistent and have varied according to several individual characteristics. For example, one study of 542 older adults reported that problem involvement and support-seeking coping were associated with lower cortisol levels ([41]).

The purpose of the current study was to examine the profiles of diurnal cortisol in a sample of older adults in terms of awakening response and slope. We then aimed to examine the concurrent associations of sociodemographic, social support, and coping styles on three indicators of diurnal cortisol disturbances.

## 2. Materials and Methods

### 2.1. Participants

The present study was a cross-sectional analysis using the baseline data from a larger randomized controlled trial of the effects of a stress management intervention among older adults in a community setting (the O’Stress Study, directed by P. Plusquellec). Recruitment took place in independent living residences for retirees and organizations serving the elderly, in Quebec, Canada. For this study, the inclusion criteria were participants (1) aged 55 years or older and (2) literate in French. After selecting the sample to calculate our outcome data (see criteria below), the analysis was conducted on 135 participants. Per cohort demographics, the participants were 88.1% female, aged 57 to 91 years (*M* = 76.26, *SD* = 6.99 years), 92.6% born in Canada, and 47.4% reported an annual income of CAD $40,000 or less. Participants provided written informed consent, and the study protocols were approved by the Human Research Ethics Committee of the University of Montreal, Canada (No. CERAS-2017-18-018-P).

### 2.2. Procedure

Recruitment for the study began in September 2018 and was completed in September 2019. We administered self-report questionnaires to collect demographic and psychosocial information. Cortisol samples were collected during the same week, with four samples per day on 2 non-consecutive days.

### 2.3. Measures

#### 2.3.1. Salivary Cortisol

Samples were collected by passive drooling (2 mL) into sterile containers (Sarstedt, tubes P/N 62.558.201). Participants were instructed to provide saliva samples on 2 non-consecutive days for a total of eight samples: upon awakening, 30 min after awakening, at 16:00, and at bedtime. The subjects were instructed to record the exact time of each collection in a diary. Saliva samples were collected from participants within 1 week. Upon collection by the research team, saliva samples were stored in freezers at −20 °C at the Center for Studies on Human Stress (www.humanstress.ca, URL accessed on 11 February 2025) until assayed using a high-sensitivity enzyme immunoassay kit (Salimetrics^®^, State College, PA, USA, Catalog No. 1-3102). Frozen samples were brought to room temperature and centrifuged at 15,000× *g* (3000 rpm) for 15 min. The detection range for this assay is 0.012–3 μg/dL. Duplicate assay results were run for each sample and averaged.

Selection of cortisol days for the analytic sample of the total sample of 171 participants, 154 returned a completed saliva kit (at least four samples from 1 day) to the research team. To ensure reliable data, several exclusion criteria were applied that were based on previous research ([35]; [53]). Participants who reported smoking were excluded from the analysis (*n* = 5). To account for atypical schedules, we excluded days on which a participant woke up before 4:00 a.m. or after 11:00 a.m. We also excluded days on which a participant collected the 30-min post-awakening sample earlier than 15 min or later than 45 min after their wake-up sample.

Three measures of diurnal cortisol were used based on the common disturbances of the HPA axis highlighted by [18] ([18]). First, the awakening cortisol response was assessed by calculating the difference between the awakening sample and the sample 30 min after awakening ([7]). Second, we calculated the cortisol area under the curve (AUC) with respect to ground to capture the total hormonal output over the course of the day ([46]). Third, we calculated the peak-to-bed slope as an indication of a flattening slope, which examines the rate of change in cortisol from the peak of the cortisol awakening response (CAR) to bedtime ([30]; [56]).

#### 2.3.2. Individual Characteristics

Age, sex, household income, years of education, living situation (e.g., living alone), comorbidities, and general health perception were assessed by questionnaire. Participants reported the total income received by their household in the past 12 months on a 6-point scale. Comorbidities were calculated from the number of unique diagnoses self-reported using a checklist and confirmed by medication use ([34]). The impact of physical health on an individual’s daily life was measured using the Physical Component Summary (PCS) of the SF-12 ([20]). The PCS considers several aspects that could be affected by physical and health problems, such as limitations in physical and social activities and usual roles, bodily pain, vitality, and general health perceptions. A higher score indicates a better health-related quality of life due to less interference of physical and/or health problems in daily life.

As control variables, participants’ body mass index (BMI; kg/m^2^) was calculated using a standardized procedure, as it may be associated with cortisol ([40]). A dichotomous Medication use variable was created to control for the effects of glucocorticoid, steroid-based medications, and antidepressant or anti-anxiety medications ([25]).

#### 2.3.3. Indicators of Subjective Stress

To reflect the influence of subjective stress on cortisol levels, we used the Perceived Stress Scale (PSS), a widely used self-report questionnaire ([8]), validated in French by [15] ([15]). Mean scores range from 0 to 4, with higher scores indicating greater stress. The reliability (Cronbach’s alpha) of the validated French version for our sample was 0.89. To provide additional information on subjective stress, major life events were assessed with the question: “Have you experienced one or more major or difficult events in the past year? If so, please briefly describe the event(s)”. The indicator represents the sum of the events reported by the participants.

#### 2.3.4. Indicators of Social Support Network and Coping Mechanisms

Four indicators of social support were assessed. Availability and satisfaction with one’s network were self-reported using the French version of the Social Support Questionnaire (SSQ-6; [48]; [50]). Availability represents the sum of people one can rely on in six hypothetical situations. Satisfaction with each of these situations is rated on a Likert scale from very dissatisfied (1) to very satisfied (6). It represents the perceived adequacy between the support received and needs and expectations. Instrumental and emotional support seeking were assessed using two subscales of the Proactive Coping Inventory (PCI) ([26]). Instrumental Support Seeking reflects the reliance on one’s social network for help in coping with stressors, including advice and general information. Emotional Support Seeking is the only (sub)scale that measures the emotional aspect of coping. This scale reflects the tendency to self-regulate one’s emotions with the help of others by identifying others’ feelings, evoking empathy, and seeking companionship. Mean scores range from 1 to 4, with higher scores indicating a greater use of these strategies.

The PCI has measured a wide range of coping strategies with an emphasis on engagement through problem-focused coping styles ([26]). Proactive coping involves autonomous goal setting with self-regulatory goal attainment cognitions and behaviors. Reflective coping describes cognitions that represent simulation and contemplation of comparative action plans to achieve better outcomes. Strategic planning measures the tendency to generate a goal-directed plan of action in which extensive tasks are broken down into manageable components. Preventive coping involves strategies that are activated in response to a perceived threat or in a state of worry. Avoidance coping measures the tendency to avoid action in a stressful situation by delaying. Mean scores range from 1 (not at all true) to 4 (completely true), with higher scores indicating greater use of these strategies. The instrument was translated into French by the first author. In our sample, the psychometric properties were satisfactory apart from the Avoidance Coping scale (three items; alpha = 0.41).

### 2.4. Statistical Analysis

Between 2.34% to 9.36% of the data were missing for the independent variables, with a low average mean of 4.7%. To handle these missing data, we used the multiple imputation method and calculated 20 datasets based on the distribution of the imputed variables and on the observed data using an IBM SPSS Statistics 25 software procedure (SPSS, Chicago, IL, USA). The number of imputed data sets was chosen to be equal to the percentage of incomplete cases ([38]; [58]). The settings in which participants were recruited, and the experimental condition of the larger study were included in the model as predictors. These 20 datasets were aggregated into one set of parameter estimates.

Normality of the distribution was assessed visually and with the Shapiro-Wilk test for each variable, with a normality range of −2.0 to 2.0. The cortisol awakening response (CAR) and the cortisol area under the curve (AUC) values were winsorized. Multiple regression analysis was used to examine the relationship between sociodemographic and psychosocial variables and the three diurnal cortisol markers. Assumptions were made prior to the analysis. Pearson correlations between the three cortisol indicators and the independent variables are listed in the Appendix A. In a first step, all independent variables were included in a linear regression model to test the contribution of the determinants affecting the hypothalamic-pituitary-adrenal axis, as illustrated by the theoretical model of aging, stress and resilience. The second step was to obtain a more parsimonious model. For this purpose, a backward strategy was used to remove, one by one, the variable in the equation with the smallest contribution (*p*-value), leaving only the variable that contributes to the model with a *p*-value less than 0.10. All statistical analyses were performed with IBM’s SPSS Statistics 25 software.

## 3. Results

As shown in Figure 1, on average, cortisol levels in this sample of older adults followed the expected diurnal pattern. The participants showed a rise in cortisol upon awakening that peaked at 30 min, and then we observed a decline until bedtime. However, the curve showed a small rise after awakening. Specifically, 38.52% of our sample achieved an increase in cortisol concentration of at least 50% after awakening.

To test the model of aging, stress and resilience, we performed standard multiple linear regressions using all independent variables selected for each cortisol outcome. After adjusting for BMI and medication use, the independent variables explained 2.7% and 1.2% of the variance of the cortisol awakening response (CAR) and peak-to-bed slope, respectively, and were not significant, *F*(20, 134) = 1.186, *p* = 0.279; *F*(20, 134) = 1.081, *p* = 0.379. Per the AUC values, the independent variables explained 8.6% of the variance and tended to reach significance, *F*(20, 134) = 1.628, *p* = 0.058. Coefficients and confident intervals are shown in Table 1.

In a second step, to obtain more parsimonious models we used a backward strategy to eliminate the variables that did not contribute to the model for each outcome. The coefficients of the final models after elimination are shown in Table 2. For the CAR, a significant model emerged with the contribution of health-related quality of life, emotional support seeking, and avoidance coping style, *F*(3, 134) = 4.03, *p* = 0.009. This model explained 6.3% of the variance. A lower use of avoidance coping strategies was significantly associated with a higher post-awakening cortisol secretion. For the area under the curve (AUC), a total of 13.8% of the variance was explained by the unique contribution of medication use, age, health-related quality of life, the experience of major life events, and proactive and avoidant coping styles, *F*(6, 128) = 4.584, *p* = 0.000. Being older, reporting a higher health-related quality of life, and reporting more recent major life events were all associated with higher levels of total diurnal cortisol. A lower use of proactive and avoidant coping strategies was also associated with higher cortisol levels. For our third outcome, peak-to-bed, the final model included two variables and explained 5.7% of the variance, and was significant, *F*(2, 134) = 5.051, *p* = 0.008. In this model, a low health-related quality of life was significantly associated with a more flattened slope.

## 4. Discussion

The aim of the present research was to examine the contribution of individual characteristics, coping styles, and social support, on three measures of diurnal cortisol in older adults. In the model of aging, stress and resilience, these associations are part of the mechanisms that lead to stress resilience and healthy outcomes in old age ([18]). The model of aging, stress and resilience was developed based on empirical data available in the scientific literature for older adults. We tested the concomitant associations between these variables in our specific sample of older adults. In our sample, the obtained model of aging, stress and resilience-based models did not explain much of the variability in diurnal cortisol levels and were not significant. Conversely, the parsimonious models explained more variance, although they showed few significant associations.

Above all, according to [12] ([12]), diurnal cortisol data can be categorized into three distinct profiles: a normative diurnal curve characterized by a robust awakening cortisol response and diurnal slope, along with lower awakening and bedtime cortisol levels (observed in 73% of their sample); an elevated curve with higher morning cortisol levels, minimal cortisol awakening response (CAR), and higher bedtime cortisol levels (found in 20% of their sample); and a flattened curve characterized by a lower CAR, lower slope, and higher evening cortisol levels (observed in 7% of their participants). Although older age was associated with a greater likelihood of fitting into the elevated diurnal curve profile, the normative diurnal curve appeared to be preserved in our sample. However, the awakening cortisol response appeared to be low. The awakening cortisol response is approximately 50% to 75% and is thought to prepare individuals to engage with the environment to face the day ahead ([17]). Less than half of our sample achieved an increase in cortisol concentration of at least 50%, suggesting lower cortisol levels upon awakening, which may be predictive of a chronic stress pathway.

None of the individual characteristics included in our study were associated with the awakening cortisol response and the rate of change from the awakening peak to bedtime. This finding is not surprising as the relationship between aging and hypothalamic-pituitary-adrenal (HPA) axis functioning has been shown to be complex. The association between the CAR and individual characteristics, usually included as covariates, is inconsistent across studies. This lack of significant association could be explained by the characteristics of our sample, which included a high proportion (43%) of individuals who were already in a state of psychological distress according to their score on the Hospital Anxiety and Depression Scale. In this context, other variables are likely to better explain the variance in CAR. For example, in their study of 156 older adults, [1] ([1]) suggested that the variance in CAR and on the cortisol diurnal slopes was better captured by the psychosocial and emotional experiences of the previous or same day. Despite this lack of associations, an older age was associated with higher levels of overall cortisol (per the AUC). The association between age and diurnal cortisol levels was consistent with changes in the HPA axis observed with aging described in previous reviews ([18]; [24]).

Our models could have included other individual variables to better explain the outcome seen in our sample. For example, studies have reported an altered CAR in the presence of depressive and anxiety symptoms in older adults ([29]; [57]). Again, the individuals in our sample had significant levels of anxiety and depression. We chose not to include these measures because we were interested in specifically testing the determinants of resilience to stress rather than health outcomes. To further explore the potential influence of psychological distress on CAR, we conducted subgroup analyses based on the presence or absence of distress symptoms, as measured by the Hospital Anxiety and Depression Scale. Although no significant difference in CAR variance was found between the two groups, the regression models revealed distinct patterns of associations within each subgroup (see Appendix B, Table A1). Among individuals with psychological distress, CAR was significantly associated with higher health-related quality of life, greater use of emotional support seeking, and strategic planning coping strategies, while greater use of reflective coping was linked to lower CAR. In contrast, for individuals without distress, higher CAR was associated with older age, better perceived health-related quality of life, greater reliance on preventive coping, and less frequent use of instrumental support seeking. These findings, although exploratory and limited by statistical power, suggest that coping mechanisms may differentially relate to physiological stress responses depending on psychological distress status. They highlight the importance of considering individual psychological profiles when examining stress markers such as CAR. Other variables, such as gender differences, could not be well examined in our study because we recruited mostly women. Sleep quality may also play a role in diurnal cortisol levels ([62]) and should be accounted for in future studies.

One of the individual characteristics of aging is the presence of age-related conditions or diseases that can affect daily functioning and, consequently, the quality of life of those affected. Age-related conditions or diseases are multifactorial, and they play a crucial role in the heterogeneity usually observed in this age group ([60]). In our sample, less interference from physical problems influenced our three measures of cortisol. We found a higher post-awakening response, higher total cortisol levels, and a higher rate of change from peak at awakening to bedtime. However, a U-shaped association rather than a linear relationship should be considered when examining CAR, as both low and high CAR have been associated with different indicators in older adults ([32]). Both elevated and low responses upon awakening could signal an HPA axis dysfunction ([18]). Given the low cortisol levels upon awakening in our sample, it is possible that a higher quality of life could indicate a healthy cortisol response. Conversely, a lower quality of life could be predictive of chronic stress, as noted above.

In aging, several measures of cortisol levels have been associated with health interferers, although they have been measured and conceptualized differently. Some studies have suggested that cortisol is a risk factor for age-related diseases ([6]; [51]). Chronic diseases usually lead to hyperactivation of the HPA axis ([6]). These consequences are illustrated by the stress theory of aging, which states that the cumulative effect of stress disrupts normal cellular function, ultimately resulting in systemic dysfunction and aging ([21]). For example, in a sample of 799 adults aged from 34 to 84 years, disruptive profiles of diurnal cortisol (elevated and flattened response) predicted functional limitations about 10 years later, measured as the interference of health problems during the performance of specific daily tasks. This prediction was stronger for individuals aged 66 years or older ([45]). Stress-related conditions and diseases generate stressors to cope with and alter the quality of life ([63]). As a natural consequence of the heterogeneity of the aging population, significant variation in cortisol levels and patterns of change are found in the literature; some people will experience an increase, decrease, or stability in cortisol levels over time ([45]). Thus, the relationships between cortisol levels, aging, disease, and conditions that disrupt one’s life need to be explored, as they may affect the maintenance of autonomy and well-being in older adults.

In our study, we measured environmental stressors with the experience of recent major life events. We found that the number of major life events in the past year contributed to a greater cortisol production throughout the day. Our result is consistent with [23] ([23]), who found higher levels of cortisol in adults aged 63 to 93 years (*n* = 1055) who reported experiencing two or more late-life events. This finding underscores the importance of helping older adults cope with stressful situations to reduce chronic cortisol exposure and prevent its detrimental effects on their physical and mental health. Indeed, stress management programs have shown small to moderate effects in reducing cortisol levels in adults and older adults ([28]; [52]).

We were unable to show any contribution of the social network determinant, and we found only a little contribution of coping styles to the interindividual variability in our three cortisol outcomes. The CAR and AUC were partially explained by the avoidance coping style, yet its consistency was poor, probably because it was the smallest instrument we used, with only three items. The avoidance scale from the PCI ([26]), designed to assess the tendency to defer action in stressful situations, has shown cultural nuances in its translation. Two items suggest a “composed restraint”, while one indicates “evasion”, aligning with the authors’ definition. Studies generally associate avoidance coping with heightened physiological stress and distress ([3]; [4]; [39]). We tentatively propose that individuals who pause before responding to stressors may exhibit lower cortisol levels, potentially indicating a form of emotional regulation. However, our research did not delve into specific emotional regulation strategies. Conversely, older adults have shown greater resilience to stress when using problem-solving strategies and social support-seeking coping strategies ([9]; [42]). Our findings did not support these results, although the proactive coping style (a problem-solving strategy) has tended to contribute to overall diurnal cortisol output.

The main limitation of this study was its cross-sectional design, which prohibits causal interpretation. Another limitation was the lack of equipment to automatically measure the time at which saliva samples were collected. Adherence to the time protocol may have contributed to the lack of associations seen. It has been shown that strict adherence to protocol is necessary to avoid spurious blunting of the CAR, as an increase was only observed in participants who adhered to the 30-min delay between the first two samples of the day in [61] ([61]). However, our participants consciously filled out their saliva collection logs.

In conclusion, our results did not support the model of aging, stress and resilience. It did show that age, health problem interferers, major life events, and avoidance coping may indeed influence diurnal cortisol in older people; but in our sample, they explained only a small part of the total interindividual variability. One reason for the lack of predictive power of our models may be the specificity of our sample, as discussed above, and the measure of stress regulation we used. Some authors have emphasized that coping style is part of a broader concept called affect regulation ([55]).

## Figures and Tables

**Figure 1 behavsci-15-00631-f001:**
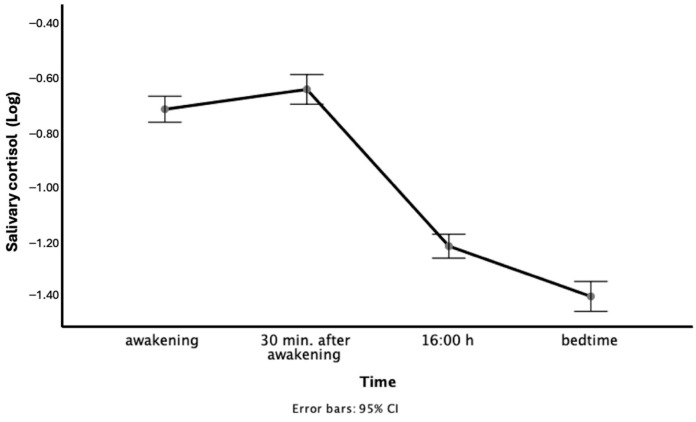
Average diurnal profile of cortisol levels in our sample.

**Table 1 behavsci-15-00631-t001:** Results of multiple linear regressions per cortisol outcomes for models including all variables.

Variables	CAR				AUC				Peak-to-Bed			
	B	*p*	95% CI	*ß*	B	*p*	95% CI	*ß*	B	*p*	95% CI	*ß*
Control												
Body mass index	0.001	0.804	−0.004, 0.005	0.024	0.026	0.123	−0.007, 0.058	0.144	0.000	0.592	0.000, 0.000	0.052
Medication use	−0.026	0.291	−0.075, 0.023	−0.105	−0.351	0.084	−0.750, 0.048	−0.168	−0.003	0.191	−0.008, 0.002	−0.132
Related to aging												
Age	0.002	0.356	−0.002, 0.005	0.095	0.034	0.022	0.005, 0.064	0.230	0.000	0.334	0.000, 0.000	0.100
Sex ^a^	0.054	0.158	−0.021, 0.129	0.141	0.219	0.482	−0.395, 0.832	0.068	0.001	0.851	−0.006, 0.008	0.019
Income	−0.007	0.451	−0.027, −0.012	−0.084	0.031	0.697	−0.128, 0.191	0.042	0.000	0.865	−0.002, 0.002	0.019
Education	0.003	0.385	−0.004, 0.011	0.091	0.017	0.606	−0.047, 0.080	0.052	0.000	0.690	−0.001, 0.001	0.042
Living situation ^b^	0.012	0.673	−0.042, 0.066	0.046	−0.222	0.321	−0.662, 0.219	−0.104	−0.001	0.579	−0.006, 0.004	−0.060
Health-related QOL	0.004	0.003	0.001, 0.006	0.307	0.029	0.006	−0.006, 0.008	0.273	0.000	0.017	0.000, 0.000	0.248
Comorbidities	0.003	0.525	−0.009, 0.015	0.054	−0.011	0.817	−0.105, 0.083	−0.023	0.000	0.808	−0.001, 0.001	−0.025
Subjective stress												
Major life events	0.004	0.465	−0.012, 0.020	0.044	0.168	0.012	0.038, 0.299	0.232	0.001	0.373	−0.001, 0.002	0.084
Perceived stress	0.011	0.632	−0.033, 0.054	0.053	0.045	0.801	−0.311, 0.402	0.027	−0.002	0.304	−0.006, 0.002	−0.115
Coping strategies												
Proactive	0.008	0.861	−0.081, 0.097	0.021	−0.595	0.108	−10.323, 0.132	−0.191	−0.008	0.051	−0.017, 0.000	−0.242
Strategic planning	0.053	0.070	−0.004, 0.111	0.215	0.011	0.963	−0.461, 0.483	0.005	0.001	0.761	−0.005, 0.006	0.036
Reflexive	−0.090	0.093	−0.194, 0.015	−0.250	−0.297	0.492	−10.152, 0.557	−0.099	−0.004	0.386	−0.014, 005	−0.129
Preventive	0.039	0.331	−0.040, 0.118	0.117	0.386	0.237	−0.257, 10.029	0.139	0.005	0.190	−0.002, 0.012	0.160
Avoidance	−0.022	0.388	−0.073, 0.029	−0.089	−0.398	0.060	−10.901, 0.060	−0.189	−0.003	0.257	−0.007, 0.002	−0.118
Social support												
Availability	−0.001	0.734	−0.003, 0.002	−0.038	−0.009	0.453	−0.034, 0.015	−0.082	0.000	0.894	0.000, 0.000	−0.015
Satisfaction	0.007	0.553	−0.016, 0.029	0.060	0.055	0.547	−0.126, 0.236	0.059	0.000	0.698	−0.002, 0.002	0.039
ESS	0.055	0.064	−0.003, 0.113	0.222	0.105	0.661	−0.369, 0.580	0.051	0.003	0.272	−0.002, 0.008	0.132
ISS	−0.038	0.267	−0.106, 0.030	−0.147	−0.115	0.682	−0.669, 0.439	−0.053	−0.001	0.682	−0.008, 0.005	−0.055

Note. ^a^ Female = 1; ^b^ living alone = 1; CAR = cortisol awakening response; AUC = area under the curve with respect to the ground; ESS = emotional support seeking; ISS = instrumental support seeking; QOL = quality of life.

**Table 2 behavsci-15-00631-t002:** Results of backward linear regressions per cortisol outcomes for the final models.

Variables	B	*p*	95% CI	*ß*
CAR				
Health-related quality of life	0.003	0.007	0.001, 0.005	0.229
Emotional support seeking	0.035	0.094	−0.006, 0.076	0.141
Avoidance coping	−0.035	0.093	−0.077, 0.006	−0.142
AUC				
Medication use	−0.342	0.055	−0.642, 0.008	−0.163
Age	0.031	0.015	0.006, 0.056	0.208
Health-related quality of life	0.026	0.004	0.009, 0.044	0.246
Major life event(s)	0.164	0.008	0.043, 0.285	0.226
Proactive coping	−0.526	0.051	−10.055, 0.002	−0.169
Avoidance coping	−0.398	0.027	−0.749, −0.047	−0.189
Peak-to-bed				
Medication use	−0.004	0.054	−0.008, 0.000	−0.162
Health-related quality of life	0.000	0.024	0.000, 0.000	0.193

Note. CAR = cortisol awakening response; AUC = area under the curve with respect to the ground.

## Data Availability

The data presented in this study are available on request from the corresponding author due to ethical reason.

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
