# Peer review of "Exploring Interindividual Variability in Resilience to Stress: Social Support, Coping Styles, and Diurnal Cortisol in Older Adults"

_behavsci, 2025, doi:10.3390/bs15050631_

Round 1
Reviewer 1 Report
Comments and Suggestions for Authors
This cross-sectional study uses baseline data from the O’Stress Study to examine the association between subjective stress and diurnal cortisol in a sample of older adults. The idea of this study is solid and takes advantage of an ongoing study, but has some inherent limitations that the authors acknowledge, specially, a high proportion of their small sample exhibiting high stress levels. The paper was nevertheless informative despite being underpowered in a few respects to address the core research questions.
Please avoid using words that may imply change over time when you are making simple comparisons. For example, in the abstract in place of “increased” please use “higher” or “greater”.
The last sentence of the first paragraph of the Introduction should include appropriate references.
The last sentence of the Introduction uses the term “concurrent effects” which implies causation. “Effects” should be replaced by “associations” or similar terminology.
The authors speculate that the absence of an association between cortisol awakening response, may be due to the high proportion of the sample with elevated depression and anxiety. This explanation begs for a subset analysis conducted within persons with lower, more normal levels of anxiety and depressive symptoms (even if underpowered) to provide a more informed discussion.
If MASR stands for model of aging, stress and resilience, then it is not correct and potentially confusing to refer to the MASR model in the discussion. In this field, is MASR a standard, accepted acronym? If not, please drop it from the paper.
Author Response
Comments 1: Please avoid using words that may imply change over time when you are making simple comparisons. For example, in the abstract in place of “increased” please use “higher” or “greater”.
Response 1: Thank you for this observation. We have revised the manuscript to remove terminology that may imply temporal change when making cross-sectional comparisons. Terms such as “increased” have been replaced with “higher” or “greater” where appropriate.
Comments 2: The last sentence of the first paragraph of the Introduction should include appropriate references.
Response 2: Thank you for pointing out this omission. We have now included the appropriate references in the final sentence of the first paragraph of the Introduction.
Comment 3: The last sentence of the Introduction uses the term “concurrent effects” which implies causation. “Effects” should be replaced by “associations” or similar terminology.
Response 3: This terminology has been corrected as suggested. The word “effects” has been replaced with “associations” to more accurately reflect the correlational nature of the findings.
Comment 4 : The authors speculate that the absence of an association between cortisol awakening response, may be due to the high proportion of the sample with elevated depression and anxiety. This explanation begs for a subset analysis conducted within persons with lower, more normal levels of anxiety and depressive symptoms (even if underpowered) to provide a more informed discussion.
Response 4:We appreciate this insightful suggestion. As recommended, we conducted additional analyses on subsamples stratified by distress levels. The results are now included in the supplementary material (Appendix B) and discussed briefly in the discussion (see line 321).
Comments 5: If MASR stands for model of aging, stress and resilience, then it is not correct and potentially confusing to refer to the MASR model in the discussion. In this field, is MASR a standard, accepted acronym? If not, please drop it from the paper.
Response 5: Thank you for this clarification. We agree that MASR is not a standard or widely accepted acronym in this field. We have revised the manuscript accordingly by removing the acronym and instead referring to the full name—the model of aging, stress, and resilience—each time it is mentioned
Reviewer 2 Report
Comments and Suggestions for Authors
This is a very interesting study of interinidividual variability in stress resilience, based on relevant theories and models including specific factors (life-events, social support, coping style, etc.). Given the relationship between chronic stress (and elevated cortisol levels) and health problems, especially in late life, elucidating factors related to resilience is important as a guide to intervention. The paper is well written and concise. My only question pertains to the sample. Given the age range, and despite the participants living in the community, I can't help but wonder whether any of them had cognitive decline (as in MCI or dementia). There is no mention of that as an exclusion criterion. Also, does the fact that some were recruited in residential homes for retirees reflect on their ability to live independently, and if so, is this due to physical disability or also signs of cognitive decline?
Author Response
Comments 1: My only question pertains to the sample. Given the age range, and despite the participants living in the community, I can't help but wonder whether any of them had cognitive decline (as in MCI or dementia). There is no mention of that as an exclusion criterion. Also, does the fact that some were recruited in residential homes for retirees reflect on their ability to live independently, and if so, is this due to physical disability or also signs of cognitive decline?
Response 1. Thank you for this thoughtful question. The broader research project was designed with both preventive and effectiveness objectives in mind. Therefore, exclusion criteria were intentionally limited in order to assess an intervention under real-world conditions. While participants with some degree of cognitive decline may have been included, the complexity of the research protocol itself likely acted as an informal filter, as individuals with more severe cognitive impairments would have been less able to comply with the study requirements. We acknowledge that including a variable related to cognitive functioning would have been relevant in this context.
Regarding the recruitment setting, participants were indeed recruited from independent living residences for retirees. These residences are designed for individuals who maintain functional autonomy and are comparable to living independently at home. In Quebec Canada, many older adults (about 17%) choose such residences for reasons including increased safety, reduced domestic responsibilities due to a smaller living space, and access to community life—not necessarily due to physical or cognitive decline. To clarify this in the manuscript, we added the following statement: “Recruitment took place in independent living residences for retirees” (see line 95).